

# Paleo-hydrologic reconstruction of 400 years of past flows at a weekly time step for major rivers of Western Canada

Andrew R. Slaughter, Saman Razavi

Global Institute for Water Security, University of Saskatchewan, Saskatoon, Saskatchewan, Canada

5 *Correspondence to*: Andrew Slaughter (andrew.slaughter@usask.ca)

25

30



**Abstract.** The assumption of stationarity in water resources no longer holds, particularly within the context of future climate change. Plausible scenarios of flows that fluctuate outside the envelope of variability of the gauging data are required to assess the robustness of water resources systems to future conditions. This study presents a novel method of generating weekly-time-step flows based on tree-ring chronology data. Specifically, this method addresses two long-standing challenges with paleo-reconstruction: (1) the typically limited predictive power of tree-ring data at the annual and sub-annual scale, and (2) the inflated short-term persistence in tree-ring time series and improper use of prewhitening. Unlike the conventional approach, this method establishes relationships between tree-ring chronologies and naturalised flow at a biennial scale to preserve persistence properties and variability of hydrological time series. Biennial flow reconstructions are further disaggregated to weekly, according to the weekly flow distribution of reference two-year instrumental periods, identified as periods with broadly similar tree-ring properties to that of every two-year paleo-period. The Saskatchewan River Basin (SaskRB), a major river in Western Canada, is selected as a study area, and weekly flows in its four major tributaries are extended back to the year 1600. The study shows that the reconstructed flows properly preserve the statistical properties of the reference flows, particularly, short- to long-term persistence and the structure of variability across time scales. An ensemble approach is presented to represent the uncertainty inherent in the statistical relationships and disaggregation method. The ensemble of reconstructed weekly flows are publically available for download from https://doi.org/10.20383/101.0139 (Slaughter & Razavi, 2019).

## 1 Introduction

Water management has traditionally assumed that variability in streamflow fluctuates within a boundary represented by the limited amount of observed data provided by gauging stations. This limited variability has been used to evaluate and manage risks to water management systems. However, it is becoming increasingly clear that the assumption of stationarity can no longer be taken for granted (Milly et al., 2008), particularly within the context of future climate change.

An opportunity exists to derive a paleo-hydrological record that will contain the effects of past climate change. Reconstructions of flow from tree-ring records for various parts of the world have shown relatively high long-term variability in hydrologic conditions at the multi-decadal to multi-century temporal scale (Agafonov et al., 2016; Axelson et al. 2009; Boucher et al. 2011; Brigode et al., 2016; Case & MacDonald 2003; Cook et al., 2004; Ferrero et al., 2015; Gangopadhyay et al., 2009; Lara et al., 2015; Maxwell et al., 2011; Mokria et al., 2018; Razavi et al. 2015, 2016; Sauchyn et al. 2011; Urrutia et al., 2011; Woodborne et al., 2015; Woodhouse & Lukus, 2006; Woodhouse et al., 2006). Therefore, tree-ring records could provide an opportunity to evaluate water resources systems against a wider envelope of hydrological variability than what is represented in the gauged records. This is critical for improving the resilience of water resources systems to future conditions.

However, there are various challenges associated with the use of tree-ring data to reconstruct streamflows that must be confronted, as outlined by Razavi et al. (2016) and Elshorbagy (2016), that add to the uncertainty of flow reconstructions, the

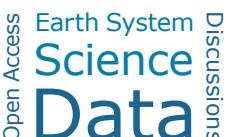

most important being: 1) tree-ring widths can only explain a portion of the variance in observed streamflow, and past studies correlating tree-ring chronologies to stream flow have obtained relatively low $R^2$ values in the range of 0.37–0.76 (Razavi et al., 2016) and; 2) there is short-term persistence in tree-ring chronologies that is typically higher than that in observed streamflow. Prewhitening techniques are commonly used to remove short-term persistence within chronology datasets.

However, Razavi & Vogel (2018) found that prewhitening can distort and reduce the structure of variability across time scales. A viable strategy of reducing the disparities in persistence between tree-ring chronologies and flow while at the same time increasing their correlative strength is to establish the statistical relationships at longer time scales of two to three years (Razavi et al., 2016; Razavi & Vogel, 2018).

Because water resources models require flow input data at daily to monthly time steps, and flow reconstructions from tree-ring data are typically at a yearly time step, some method of flow disaggregation is required. Although many studies have established relationships between tree-ring chronologies and flow, very few published studies have attempted to further disaggregate reconstructed flow. Sauchyn and Ilich (2017) estimated 900 years of weekly flows for the North Saskatchewan at Edmonton and for the South Saskatchewan at Medicine Hat. They determined the statistical relationships between tree-ring

chronologies and mean annual naturalised flow using standard dendrohydrological techniques, but further disaggregated yearly flows to weekly using stochastic downscaling while constraining the resulting weekly flows by the statistical properties of the historical record.

The present study presents appropriate approaches and tools for reconstructing flows from tree-ring data and disaggregating

yearly flows to weekly while taking into account uncertainty through an ensemble approach. Four hundred years of weekly flows were generated for the four major sub-basins of the Saskatchewan River Basin, a river of great ecological, social and economic importance in Western Canada. The approach of reconstructing flow from tree-ring chronologies contains many uncertainties, including choice of predictor chronologies and the choice of disaggregation technique. The present study presents an ensemble approach of encompassing these uncertainties.

**2 Case study, data and methodology**

**2.1 Case study**

The Saskatchewan River Basin (SaskRB; ~400,000 km$^2$) in Western Canada is a large river basin that transcends the Alberta, Saskatchewan and Manitoba provinces, and also extends into a small part of the American State of Montana (Fig. 1). The east-facing Rocky Mountains to the west at over 3,000 m in elevation act as the 'water tower' of the basin (Martz et al., 2007;

Pomeroy et al., 2005), contributing up to 95% of the total basin flow, after which the elevation of the basin drops to 750 m, 450 m and 300 m on the plains of Alberta, Saskatchewan and Manitoba, respectively, transitioning from alpine forest to prairie grasslands and river valley forest. Long sunny winters and short hot summers characterise the climate of the basin, and there



is a dramatic rainfall gradient from west to east, with precipitation of up to 1,500 mm year$^{-1}$ in the mountains to 300 mm year$^{-1}$–500 mm year$^{-1}$ on the semi-arid plains.

## 2.2 Data

Tree-ring data for the major headwater tributaries of the SaskRB were used, namely the North Saskatchewan, Red Deer, Bow
5   and Oldman rivers. Naturalised flows generated by Alberta Environment and Sustainable Resource Development for four gauging stations (1912–2001) representing each of the four headwater tributaries were used in the analysis (see Fig. 1). A total of 16 tree-ring chronology sites were used, with most from sites chosen for soil moisture availability only during snowmelt or rainfall. Razavi et al. (2016) describe the tree-ring measurement, detrending and averaging procedures used. The tree-ring chronologies used represent tree growth rates from 1600 to 2001.



Figure 1. Map of the tributaries of the Saskatchewan River Basin and the locations of the chronology sites (circles) and streamflow gauges (cones)

### 2.3 Methodology

#### 2.3.1 Reconstruction of flows based on tree-ring chronologies

The traditional method of reconstructing streamflow based on tree-ring chronologies is through correlations between tree-ring chronologies and observed streamflow at the yearly scale. However, as shown by Razavi et al. (2016), there are stronger correlations between tree-ring chronologies and streamflow at multi-year time scales. In addition, establishing correlations at multi-year time scales can be a viable method of overcoming the significantly higher persistence in tree-ring chronologies compared to streamflow without resorting to prewhitening techniques, which have been shown to result in a loss of information

not related to autocorrelation (Razavi et al., 2016; Razavi and Vogel, 2017). Although the relationship between streamflow and tree-ring chronologies strengthens at multi-year time scales, establishing this relationship at an optimal timescale that can be in the order of 5 years (Razavi et al., 2016) would disadvantage the process of disaggregation of deconstructed flow. Therefore, as a compromise, the present study established statistical relationships between two-year moving averages of tree-ring chronologies and naturalised streamflows. Multiple linear regression (MLR) fitted by least squares was the statistical

approach taken to establish the relationships between tree-ring chronologies and streamflows. The predictive ability of the models was assessed through the coefficient of determination, $R^2$. Similar to Razavi et al. (2016), only the four and eight chronology sites falling within the sub-basins of the North Saskatchewan River sub-basin and Oldman River sub-basin, respectively were used to establish their respective MLR models, whereas all 16 chronologies were used to establish MLR models for the Red Deer River and Bow River sub-basins (Fig. 1). Building on the experience of Razavi et al. (2016), models

with both three and two chronology predictors were established for the North Saskatchewan River and Oldman River sub-basins, whereas MLR models with only two chronology predictors were established for the Bow River and Red Deer River sub-basins. MLR models were generated for the shared period between tree-ring chronologies and naturalised streamflows of 1912–2001. A leave-one-out cross-validation strategy was used to test the performance of each model. To account for the uncertainty in streamflow reconstructions, multiple MLR models with the best performances were selected for each sub-basin.

The choice of the best models for each sub-basin was rather subjective, and depended on relative $R^2$ values achieved for all the models generated.

#### 2.3.2 Disaggregation of two-year reconstructed flows to weekly

Although there are definite advantages in reconstructing streamflow from tree-ring chronologies at multi-year time scales (Razavi et al., 2016), the usefulness of these flows for evaluating water resources systems is limited. Generating weekly

reconstructed streamflow would be of more use in evaluating water resources systems as, for example, the established water management model, the Water Resources Management Model (WRMM) (Alberta Environment, 2002) used within the Prairie



Provinces, runs on a weekly time step. The conceptual approach taken can be represented in Fig. 2. The basic premise adopted is that biennial reconstructed flow can be disaggregated to weekly reconstructed flow by the selection of biennial flow periods from the reference naturalised flow (1912–2001) with similar attributes to the biennial reconstructed flow, after which the weekly flow distribution of the selected reference flow period can be used to construct the weekly flows. The attributes used

to match the biennial reconstructed flow with biennial reference flow were hydrological condition, simply defined as dry, normal and wet conditions corresponding to flow less than the 25th percentile, between the 25th percentile and the 75th percentile, and greater than the 75th percentile, respectively, and which year in each biennial flow, year 1 or year 2, contributes the greater amount of flow. In Fig. 2, (1) represents the average tree-ring growth rates of all the tree-ring chronologies used to generate a particular MLR model for a sub-basin on a yearly time step. This allows B for each biennial flow in (2) to be set to

1 or 2 to indicate whether the first or second year of that biennial flow value contributed the greater flow. 'A' in (2) represents the hydrological condition explained earlier. A similar process is performed for the weekly naturalised reference flow in (3), except that average yearly flows are used to set 'B'. The biennial reconstructed flow is then stepped through in (4), and a similar period according to A and B is randomly selected from the biennial reference flow. The approach of random matching allows an ensemble of weekly flow reconstructions to be generated for each single biennial flow reconstruction. The weekly

distribution of flows in the selected biennial reference flow period is then used to construct the weekly flow reconstruction, scaled to have the same biennial average as the original reconstructed biennial flow.



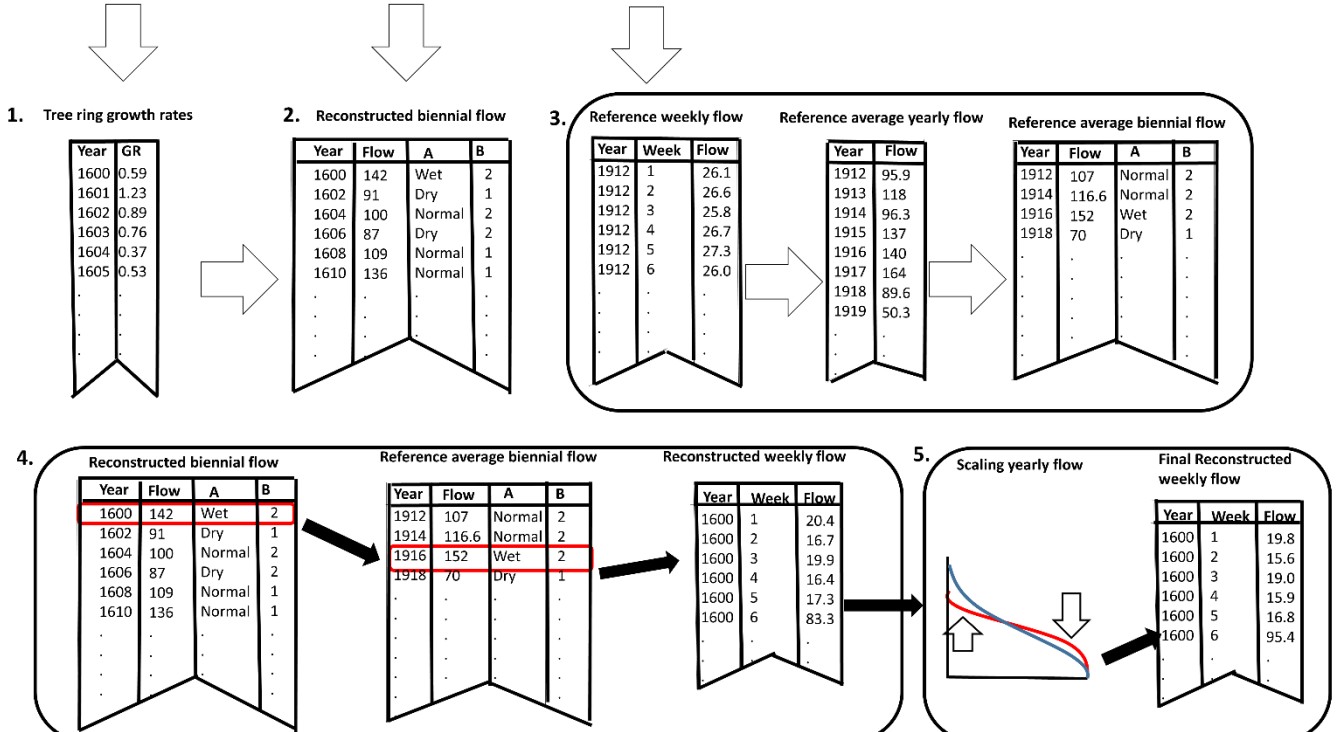

**(1)** Read in the average yearly tree-ring growth rates (GRs) used to generate the MLR model of reconstructed biennial flow.

**(2)** Read in the biennial reconstructed flow generated through the MLR model and set A and B, where A = 'Dry', 'Normal' or 'Wet' if the flow is less than the 25th percentile, between the 25th percentile and the 75th percentile, and greater than the 75th percentile, respectively, and B = 1 or 2 according to the greater of the corresponding annual GRs from **(1)**, e.g. for 1600 and 1601 in **(1)**, 1601 has the larger growth rate, therefore B is set to 2 for 1600 in **(2)**.

**(3)** Read in the weekly naturalised flow and generate a second time series of yearly average flow. Then generate a third time series of biennial average flow. Set A and B, where A = 'Dry', 'Normal' or 'Wet' if the flow is less than the 25th percentile, between the 25th percentile and the 75th percentile, and greater than the 75th percentile, respectively, and B = 1 or 2 according to the greater of corresponding average yearly flow.

**(4)** Step through reconstructed biennial flow and **randomly** identify a reference average biennial flow with the same values for A and B. Reconstruct the weekly flow using the weekly distribution of the reference weekly flow for the corresponding years.

**(5)** The disaggregation process can result in some loss of variation in the flows at a yearly scale. A scaling process was implemented to scale the flow duration curve (FDC) of the of the reconstructed yearly flow to have the same shape as that of the reference yearly flow. Statistical scaling was implemented to ensure the yearly reconstructed flows have the same standard deviation as the reference yearly flows.

Figure 2. Conceptual representation of the process for disaggregating two-year (biennial) tree-ring reconstructed streamflow to weekly reconstructed streamflow

The biennial reconstructed time series naturally demonstrate smaller variability compared with the biennial flows in the reference period, when MLR models are used for reconstruction. Therefore, the resulting annual and weekly time series also





have less variability compared with their counterparts in the reference period. To rectify this problem, the reconstructed flows generated by stage (4) in Fig. 2 over the reference period (1912–2001) were compared to the reference flow at a yearly scale in the form of flow duration curves (FDCs). Typically, loss of variance in the reconstructed flows will manifest as fewer extreme high and low flows. A scaling equation was implemented to scale the FDC of the reconstructed flow to have the same

shape as that of the reference flow:

$$q' = q \times (A \times P^B + C),$$ (Eq. 1)

where $q'$ is the scaled yearly reconstructed flow (m$^3$ s$^{-1}$), $q$ is the yearly reconstructed flow (m$^3$ s$^{-1}$), $P$ is the frequency (%) and $A$, $B$ and $C$ are parameters that are calibrated by fitting the scaled yearly flow reconstructions for the reference period (1912–2001) to the yearly reference flow FDC.

In addition, the scaled yearly reconstructed flows were re-scaled according to the mean and standard deviation of the yearly reference flows:

$$q'' = \frac{q' - q'_{stdev}}{q'_{mean}} \times Q_{stdev} + Q_{mean},$$ (Eq. 2)

where $q'_{stdev}$ and $q'_{mean}$ are the standard deviation and mean of the scaled yearly reconstructed flow for the reference period (1912–2001), respectively, $Q_{stdev}$ and $Q_{mean}$ are the standard deviation and mean of the yearly reference flow, respectively and

$q''$ is the final (re-scaled) yearly reconstructed flow for the entire reconstruction period (1600–2001).

This update of the yearly reconstructed flows is used to scale the weekly flow reconstructions, which are in turn scaled to have the same biennial average as the original biennial flow reconstructions.

## 3 Results

### 3.1 Reconstruction of biennial flows based on tree-ring chronologies

MLR models with the best $R^2$ values for each sub-basin were chosen to reconstruct biennial flows based on tree-ring chronologies (Table 1). Four, nine, six and ten MLR models were chosen for the North Saskatchewan, Oldman, Red Deer and Bow sub-basins, respectively, with $R^2$ values ranging from 0.50–0.56, 0.44–0.51, 0.45–0.55 and 0.49–0.56, respectively. Figure 3 shows the time series of reconstructed two-year (biennial) flows for all the MLR models shown in Table 1 for the

four sub-basins along with the reference flow over the calibration period.




Table 1. Regression equations and $R^2$ values obtained in two-year (biennial) flow reconstructions using tree-ring chronologies for the Saskatchewan River Basin

| Sub-basin | $R^2$ | Multiple linear regression equation |
|---|---|---|
| North Saskatchewan | 0.51 | $20.30 \times SFR + 71.33 \times DEA + 123.16$ |
| | 0.56 | $40.42 \times WPP + 64.04 \times DEA + 112.48$ |
| | 0.50 | $83.62 \times DEA + 12.75 \times TWO + 121.26$ |
| | 0.52 | $21.61 \times SFR + 62.09 \times DEA + 14.66 \times TWO + 116.74$ |
| Oldman | 0.47 | $-39.24 \times BDC + 93.32 \times CAB + 66.06$ |
| | 0.46 | $32.13 \times WSC + 55.14 \times HEM + 29.96$ |
| | 0.44 | $64.79 \times CAB + -18.03 \times ELK + 68.11$ |
| | 0.49 | $24.04 \times OMR + -52.63 \times BDC + 86.24 \times CAB + 64.02$ |
| | 0.49 | $-51.66 \times BDC + 31.88 \times WSC + 75.79 \times CAB + 65.28$ |
| | 0.51 | $-38.64 \times BDC + 71.69 \times CAB + 42.19 \times HEM + 45.61$ |
| | 0.48 | $31.64 \times WSC + 70.46 \times HEM + -33.73 \times ELK + 45.80$ |
| | 0.51 | $40.40 \times CAB + 58.02 \times HEM + -38.00 \times ELK + 53.57$ |
| | 0.49 | $42.32 \times CAB + 43.94 \times HEM + -17.24 \times BZR + 48.39$ |
| Red Deer | 0.55 | $-9.13 \times SFR + 35.68 \times WCH + 29.50$ |
| | 0.45 | $-7.16 \times WPP + 44.086 \times OMR + 24.74$ |
| | 0.46 | $-4.40 \times WPP + 39.63 \times WSC + 26.43$ |
| | 0.50 | $-3.24 \times WPP + 39.45 \times WCH + 23.41$ |
| | 0.46 | $-14.82 \times WPP + 37.04 \times JOLA + 37.81$ |
| | 0.47 | $-0.001 \times DEA + 37.39 \times WCH + 23.05$ |
| Bow | 0.53 | $31.49 \times SFR + 40.42 \times WSC + 54.93$ |
| | 0.50 | $33.42 \times SFR + 36.59 \times CAB + 56.74$ |
| | 0.50 | $36.67 \times SFR + 53.75 \times HEM + 36.18$ |
| | 0.51 | $40.40 \times SFR + 28.60 \times WCH + 54.02$ |
| | 0.49 | $26.48 \times SFR + 56.23 \times JOLA + 42.07$ |
| | 0.50 | $34.49 \times SFR + 35.19 \times LEE + 56.20$ |
| | 0.56 | $41.08 \times WPP + 39.93 \times WSC + 49.15$ |
| | 0.53 | $42.38 \times WPP + 35.25 \times CAB + 52.53$ |
| | 0.50 | $44.72 \times WPP + 49.00 \times HEM + 36.40$ |
| | 0.55 | $37.33 \times WPP + 56.48 \times JOLA + 33.87$ |





Figure 3. Time series of reconstructed 2-year moving average (biennial) flows in (a) North Saskatchewan, (b) Oldman, (c) Red

Deer and (d) Bow rivers. The shown reconstructed flows for the calibration period are the results of cross-validation



Figure 3 shows a relatively narrow distribution of reconstructed flows for the North Saskatchewan and Oldman sub-basins relative to those of the Red Deer and Bow River sub-basins, indicating a higher degree of uncertainty in the reconstructed flow for the latter pair of sub-basins. This could be related to the fact that the Red Deer and Bow River sub-basins contained few or none of the 16 tree-ring chronologies used in the biennial flow reconstructions.

5 **3.2 Disaggregation of two-year (biennial) reconstructed flows to weekly**

Figure 4 shows an example of how the scaling equation was derived for step (5) in Fig. 2 of the disaggregation process. This example is for the Bow River for one of the MLR models, and shows the FDCs of the yearly reconstructed flow up to step (4) and the yearly reference flow. The differences in the shapes of the FDCs illustrate some loss of variation in the reconstructed flows, with fewer extreme (high and low) flows. The values of *A*, *B* and *C* in Equation 1 were changed to obtain a scaling of

10 the reconstructed flow FDC to take a similar shape to that of the reference flow.

Flow (m³ s⁻¹) vs % Time equaled or exceeded

Legend: Reference — Reconstructed — Scaled




Figure 4. An example of a comparison between a yearly flow reconstruction and yearly reference naturalised flow for the reference period (1912–2001) for the Bow River using a flow duration curve (FDC). A scaling method was used to scale the yearly flow reconstruction FDC to have the same shape as the FDC of the yearly reference flow

5    For the present study, an ensemble of 30 weekly flow reconstructions for a single biennial flow reconstruction for each sub-basin was generated to illustrate the method. Fig. 5 shows the ensemble time series of the weekly flow reconstructions in relation to the reference weekly naturalised flow for a short window of the reference period for the Oldman River. It is evident that the reconstructed flows have the correct timing and similar range of flows to the reference flow, and also respond to periods of increased and decreased flow. However, further analysis was required to determine if the reconstructed flows have

10    similar statistical properties to the reference flows.

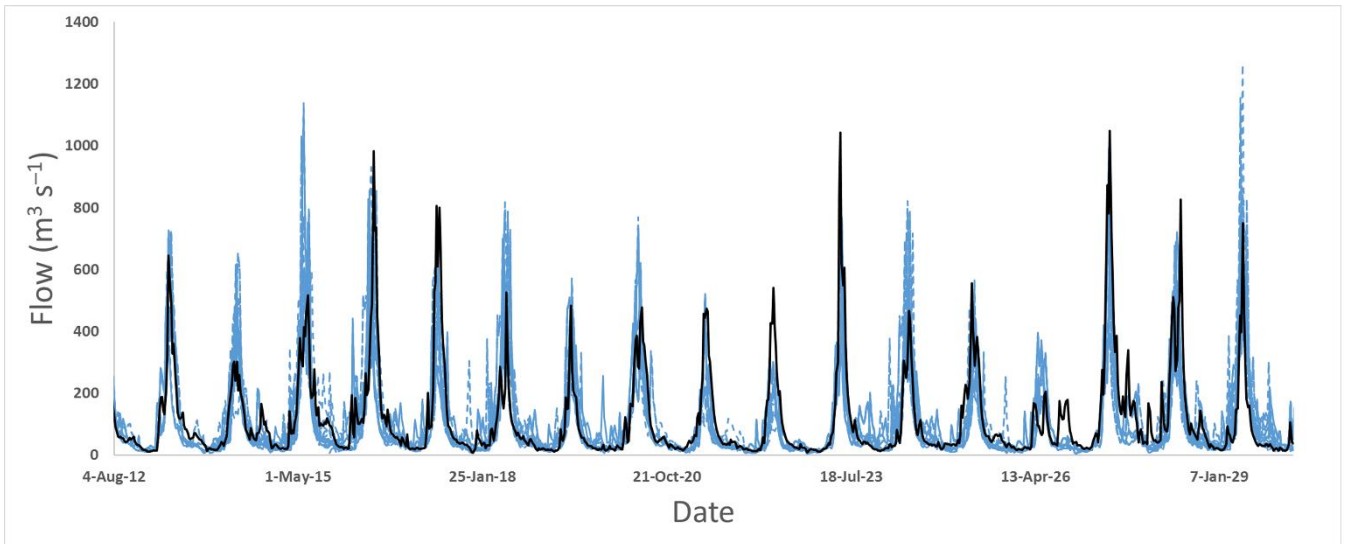

Figure 5. A time series comparison of an ensemble of 30 weekly flow reconstructions (dotted blue) and weekly naturalised flow (solid black) for the Oldman River basin.

Figure 6 shows the cumulative frequency distributions (CFDs) for the reconstructed flows in relation to the reference flows for a yearly time step, for both the reference period (1912–2001) and the full reconstruction period (1600–2001). The minimum and maximum bounds of the CFDs of the 30 reconstructed flows for the reference and full period are represented by the red dotted and blue dotted lines, respectively, whereas the CFD for the reference flow is indicated by the solid black line.

It is evident from Fig. 6 that for each sub-basin, the bounds of the CFDs for reconstructed flows for the reference period encompass the CFD of the reference flow. The bounds of the CFDs for reconstructed flows for the full period show some shifts in some cases compared to the reference flow. For example, the reconstructed flows for the Oldman River for the full period





appear to be slightly higher than those of the reference period (Fig. 6b) whereas the reconstructed flows for the North Saskatchewan River over the full period may have a smaller proportion of high flows compared to the reference period (Fig. 6a). It is important to consider that this example considers only one reconstruction model for each sub-basin, and Fig. 3 shows a fairly high variability between reconstructions for some sub-basins; therefore, the final set of flow reconstructions will contain considerably higher variability.

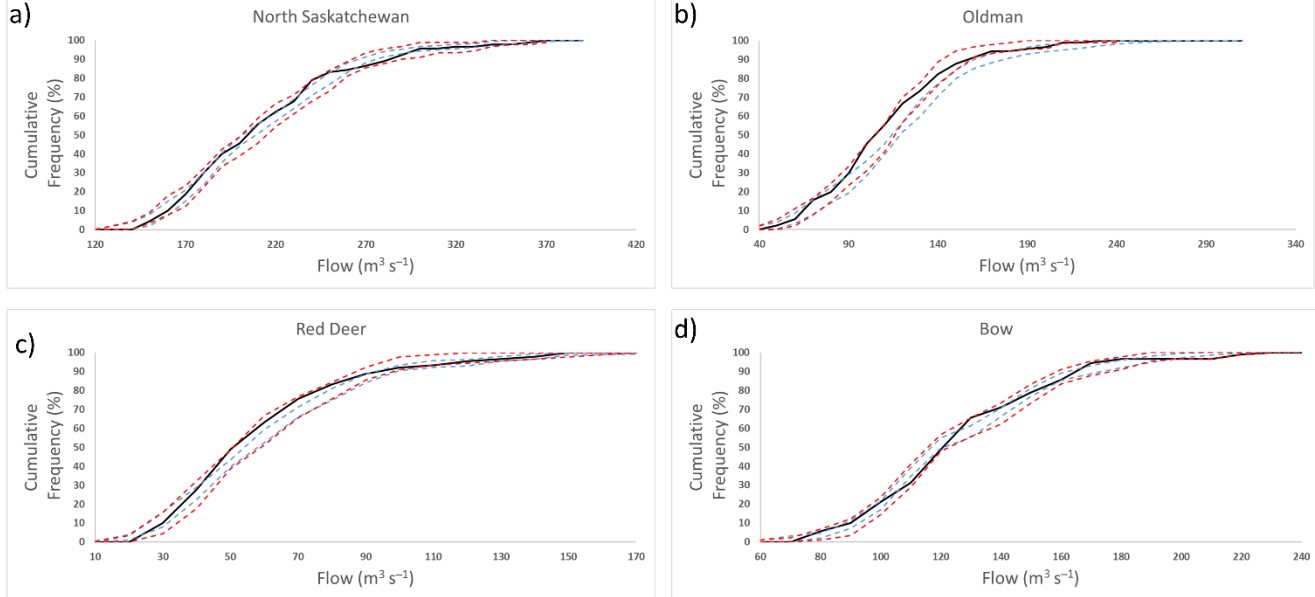

Figure 6. Cumulative frequency distributions (CFDs) between reconstructed flow and reference naturalised flow at a yearly time step. The minimum and maximum bounds of the CFDs of the 30 reconstructed flows for the reference and full period are represented by the red dotted and blue dotted lines, respectively, whereas the CFD for the reference flow is indicated by the solid black line. (a) North Saskatchewan River, (b) Oldman River, (c) Red Deer River, (d) Bow River.

Figure 7 shows a comparison of autocorrelation for reconstructed and reference flows for the reference period 1912–2001 at a yearly time step for each sub-basin, where the dotted red lines indicate the maximum and minimum of the autocorrelation values of the 30 flow reconstructions and the solid black line is the autocorrelation of the reference flow. It is evident in Fig. 7 for all sub-basins for both the reconstructed and reference flows, that for each sub-basin, the autocorrelations of the reconstructed flows are generally comparable with that of the reference flow.





Figure 7. A comparison of autocorrelation between the 30 reconstructed flows and reference naturalised flow at a yearly time step over the reference period (1912–2001), where the red dotted lines indicate the maximum and minimum bounds of the autocorrelation values of the 30 reconstructed flows and the solid black line indicates the autocorrelation of the reference flow.

5   (a) North Saskatchewan River, (b) Oldman River, (c) Red Deer River, (d) Bow River.

## 4 Discussion

Biennial reconstructions achieved for the four sub-basins in the present study were generally broadly similar to five-year reconstructions obtained for the same sub-basins by Razavi et al. (2016) using the same set of tree-ring chronologies. The major differences between the results of the present study and those of Razavi et al. (2016) are higher variation in the

10  reconstructed flows and greater divergence between the reconstructed flows for individual sub-basins for the present study





(Fig. 3). This can be related to the fact that Razavi et al. (2016) found a stronger relationship between flow and tree-ring chronologies on a five-year time step whereas the present study used a two-year time step. The $R^2$ values achieved for five-year flow reconstructions by Razavi et al. (2016) were in addition higher, ranging from 0.54 to 0.72, whereas $R^2$ values achieved in the present study ranged from 0.44 to 0.56. The present study established the relationship between flow and tree-ring chronologies on a two-year time step as a compromise to more easily facilitate the disaggregation of flows while still achieving a relatively strong relationship between flows and tree-ring chronologies and overcoming the discrepancies in persistence between flow and tree-ring chronologies without resorting to prewhitening techniques.

The approaches of using the weekly distribution of the reference flow within the disaggregation of biennial reconstructed flows [see Fig. 2 step (4)] along with the scaling of the yearly reconstructed flow [see Fig. 2 step (5)] appear to resolve the issue of discrepancies in variation and persistence between the reconstructed and reference flow. Figure 5 shows that the weekly reconstructed flow displays the same timing and range of flow in comparison to the reference flow, and also similar timing of dry and wet periods. Figure 5 and Fig. 6 show that the disaggregation approach used was successful in replicating the variance in the reference flow within the reconstructed flow. The persistence between flows for both the reference and reconstructed flows were similar, and both showed a generally decreasing trend with increasing time lag (Fig. 7).

Razavi et al. (2016) showed that the higher persistence in tree-ring chronologies compared to flow can be transferred to reconstructed flow if the relationship between tree-ring chronologies and flow is established at shorter time periods of a couple of years. However, the reconstructed flows and tree-ring chronologies show more consistent persistence properties at longer time scales. Figure 8 shows the variance of the reconstructed and reference flows at different time scales on a log–log scale for the four sub-basins. The variance values at different time scales were calculated through averaging, so for example, a flow period of 100 years would yield 50 and ten values when the average of every two and ten years is calculated, respectively. The graph represents variance of the different time series over different time scales. The slopes of the different time series can be benchmarked against a random process (the red dotted line in the plot) which contains no persistence at any time scale. The differences in slopes between the reconstructed and reference flows shown in Fig. 8 compared to that of the random process can be attributed to persistence at the range of time scales represented. Razavi et al. (2016) using a similar plot showed that tree-ring growth rates have considerably different persistence at shorter time scales compared to flow, and this persistence could be expected to be transferred to flow reconstructed from tree-ring chronologies using relationships established at shorter time scales. The slope associated with reference flow would however be closer to that of the random process at a shorter time scale. Figure 8 demonstrates that the flow reconstruction and disaggregation method used in the present study appears to overcome the problem of transferal of higher persistence in tree-ring chronologies to the reconstructed flow at shorter time scales. The slopes of both the reference and reconstructed flows appear similar to the random process, represented by the straight red line, at shorter time scales, after which their respective slopes follow similar trajectories as they diverge from that of the random process. This indicates that the reconstructed flows have similar persistence to the reference flows across the



entire reference flow period of 90 years. Variance versus time scale plots provide a method of studying the 'Hurst Phenomenon' for long-term persistence in hydrologic time series (Hurst, 1951).

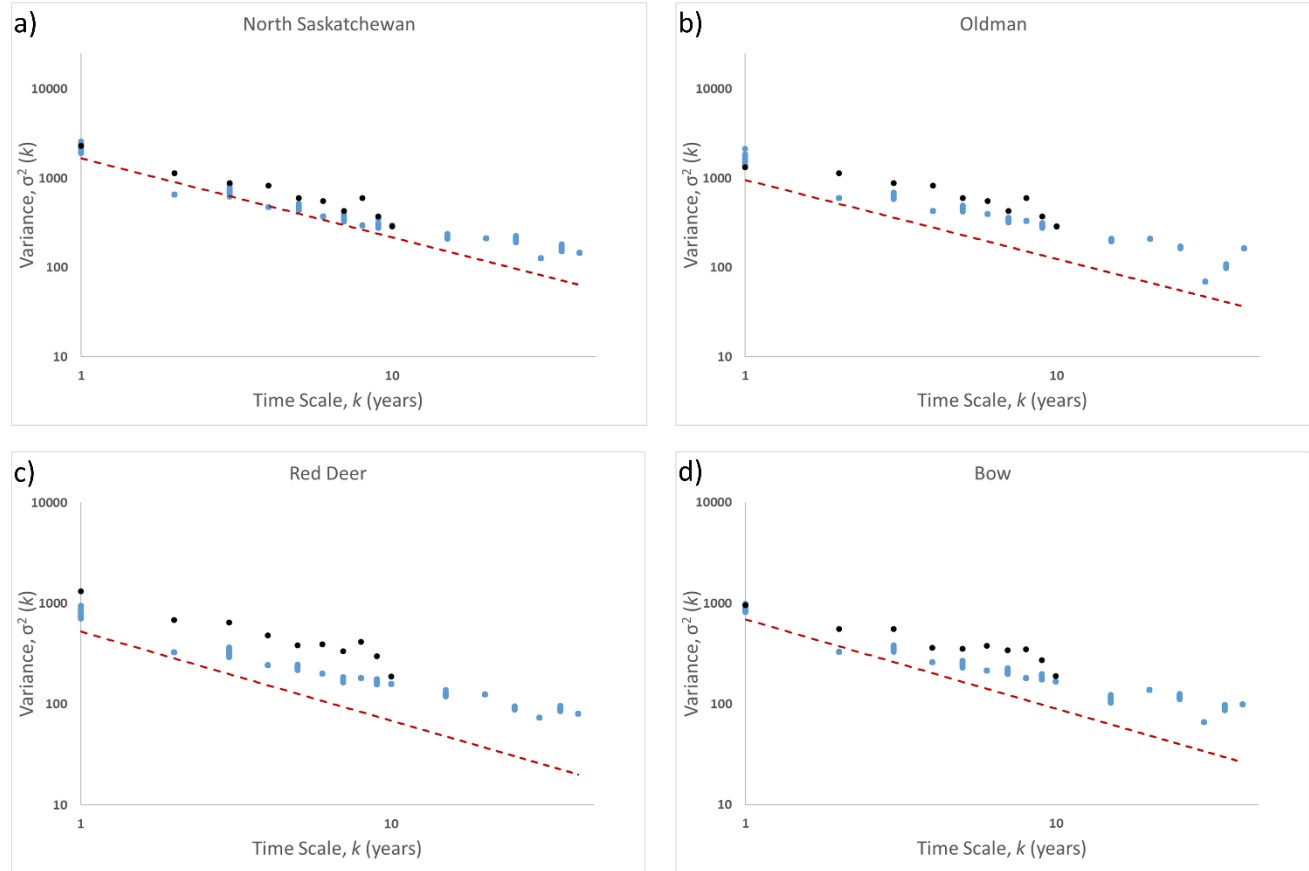

Figure 8. Variance-versus-time-scale plot for reference and reconstructed flows in the Bow River. Black and blue time series represent reference and reconstructed flows, respectively, whereas the red line represents a random process with no long-term persistence.

In the present study, the reconstructions of biennial flow for the North Saskatchewan and Oldman River sub-basins showed lower divergence compared to those for the Red Deer and Bow River sub-basins (Fig. 3). This indicates a higher degree of uncertainty in the flow reconstructions for the Red Deer and Bow River sub-basins, and is due to the fact that few or none of the 16 tree-ring chronologies used occurred in these sub-basins. There are many additional uncertainties in the statistical relationships constructed between flow and tree-ring chronologies (Razavi et al., 2016). These uncertainties are amplified by the further disaggregation of biennial flow reconstructions to weekly. An approach to represent this uncertainty is to generate



a large reconstructed flow ensemble. Uncertainty within the statistical relationship between flow and tree-ring chronologies can be represented by using multiple acceptable MLR models. For each biennial reconstructed flow generated by a single MLR model, the uncertainty in the disaggregation process can be represented by generating a large number of weekly reconstructed flows. In the present study, the number of weekly reconstructed flows generated for each acceptable MLR model was chosen

to obtain an ensemble of 500 weekly flow time series for each sub-basin. To demonstrate the full range of uncertainty of the reconstructed flows, Fig. 9 shows the yearly average flow range ($5^{th}$ and $95^{th}$ percentiles) of the 500 time series of weekly reconstructed flows for each sub-basin. It is evident that the higher uncertainty between reconstructed biennial flows, as represented in Fig. 3, results in higher uncertainty in the weekly flow reconstructions, as represented in Fig. 9. For example, there are relatively little differences between the biennial flow reconstructions for the North Saskatchewan River (Fig. 3a), and

this results in relatively smaller uncertainty in the weekly flow reconstructions (Fig. 9a), whereas the large uncertainties in the biennial flow reconstructions for the Red Deer (Fig. 3c) translate into larger uncertainty in the corresponding weekly flow reconstructions (Fig. 9c). In general, Fig. 9 shows that there is a large uncertainty range in the weekly flow reconstructions. Reducing the total uncertainty in the weekly flow reconstructions therefore requires establishing stronger statistical relationships between the reference biennial flow and biennial tree ring chronologies for each sub-basin. If new tree-ring

chronology data falling within the Red Deer River and Bow River catchments become available, future studies could attempt to establish stronger statistical relationships with the reference flow for these two sub-basins.

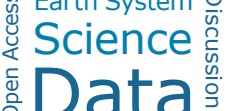



Figure 9. Time series of reconstructed yearly average flows in (a) North Saskatchewan, (b) Oldman, (c) Red Deer and (d) Bow rivers showing the 5th and 95th percentile range of the ensemble of 500 time series flows as a representation of uncertainty for 1600–2001 (dotted blue lines) as well as the observed reference flow (solid black line).



## 6 Data availability

## 7 Conclusions

The present study presented a novel method for generating weekly flows for a period preceding the period of record, based on
tree-ring data, while maintaining the statistical properties of the reference (measured) flows, including variability and
persistence across all time scales. The method featured two novel components; unlike the conventional approach that mainly
bases the analysis on annual (or sub-annual) flow-chronology correlations, the method (I) first reconstructs flows on a biennial
(two-year) scale which demonstrates higher correlation with chronologies (tree growth), thereby resulting in a higher
explanatory power, and (II) then disaggregates the biennial flows into annual and weekly scales based on information contained
in the annual chronologies and weekly flow data in the reference period. The weekly reconstructed flows for the Saskatchewan
River Basin, a large river basin in Western Canada which is of great social and economic importance, facilitates the
investigation of multiple flow futures, which can contribute to increasing the resilience of the basin to future climatic changes.

## Author contribution

SR provided supervision for all aspects of the study and critically evaluated the methods and results as well as the manuscript.
SR specified the statistical techniques used, a broad concept of the disaggregation technique and methods to validate the results.
AS conducted all analyses, wrote the code for the disaggregation technique and prepared the manuscript.

## Competing interests

The authors declare that they have no conflict of interest.

## Special issue statement

This is a contribution to the special issue on water, ecosystem, cryosphere, and climate data from the interior of Western
Canada and other cold regions

## Acknowledgements

The authors wish to acknowledge David Sauchyn for generating the tree-ring data used in this paper. This study was funded
by the Integrated Modelling Program for Canada (IMPC) and the Global Water Futures (GWF). The tree-ring data used in this



paper are archived in the International Tree-Ring Data Bank (ITRDB), available online at http://www.ncdc.noaa.gov/data-access/paleoclimatology-data/datasets/tree-ring.

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
