# Peer review of "Paleo-hydrologic reconstruction of 400 years of past flows at a weekly time step for major rivers of Western Canada"

_Earth System Science Data, 2019_

## Referee Comment (RC1) · Anonymous Referee #1 · 11 Oct 2019

The tree-ring data are used to reconstruct flow data in the Saskatchewan River Basin with a weekly time step over the past 400 years. The datasets and the methodology are valuable to the ESSD research community. The archived data is consistent with the manuscript and well organized with a readme that helps the users to efficiently use the data. Therefore, I recommend it to ESSD; however, I believe that addressing the following comments can clarify the methodology and increase its reproducibility of the work. 1. Arguably, as mentioned in the abstract, "Plausible scenarios of flows that fluctuate outside the envelope of variability of the gauging data are required to assess the

robustness of water resources systems to future conditions." The question that remains unanswered is: why should we make sure that the reconstructed flows "properly preserve the statistical properties of the reference flows, particularly, short- to long-term persistence and the structure of variability across time scales", mentioned later in the abstract and used in the methodology. 2. Page 5: "Similar to Razavi et al. (2016), only the four ..." needs to be explained more in-depth in this manuscript to make it self-contained. 3. Adding to comment 2 above, the following methods/approaches need to be explained in Section 2 to make the manuscript self-contained: a. Page 5: why MLR is expected to reconstruct flow from tree ring data, adequately. b. Page 5: The "leave-one-out cross-validation strategy" c. Page 6: The "random matching" 4. Authors need to elaborate on the negative regression coefficients in table 1. Are they physically meaningful? Could the regression be constrained to take only non-negative coefficients? In the same table, variables (e.g. WWP and JOLA) have to be introduced. 5. Page 7: What do author mean by naturally in "The biennial reconstructed time series naturally demonstrate smaller variability compared with the biennial flows in the reference period, when MLR models are used for reconstruction." 6. The two-year instrumental periods briefly introduced in the abstract need to be explained more in-depth in Section 2. 7. The persistence calculation should be explained in Section 2.

---

## Referee Comment (RC2) · Anonymous Referee #2 · 18 Oct 2019

Using a statistical approach (multiple linear regression and a disaggregation framework) the study generates reconstructed flows at weekly temporal scale based on tree-ring chronology data. This approach is shown to retain the variation and persistence of the observed flows of the Saskatchewan River Basin over the reference period of 1912-2001. I thought this was a concise paper, which presents the methodology and data clearly. I recommend publication after the following comments are addressed.

Specific comments: 1. The authors use $R^2$ to select the best MLR model. Did they consider possible over-parameterization and multicollinearity? Analyzing the standard

errors and metrics such as information criteria (e.g. AIC) can identify the optimal models more reliably.

2. It is mentioned that "weekly flow distribution of the selected reference flow period can be used to construct the weekly flows". By doing so, the variability in the reconstructed flow will be similar to the reference flow variability, correct? Please discuss how/if this assumption can undermine the estimated variability of reconstructed flows? How does it account for possible recent trends due to the anthropogenic climate change effects?

3. The study focuses on matching the variation and persistence between the observed flow in the reference period and the reconstructed flow. I wonder whether historical/recent physical processes could have distorted this similarity? For example, recent climate change trends are much stronger compared to the historical periods (prior to the reference period)

4. P5/L8: I suggest discussing briefly the cause(s) for persistence in tree-ring chronologies and based on that justify why the multi-year approach will overcome the problem. Related to this, how about the role of teleconnection signals, which can affect the records over multiple years.

5. P6/L15-16- It matches the first moment (i.e. the mean). How about higher moments like the variance? 6. Figure 2, step 3- I wonder how much the yearly average is affected by seasonal variations? i.e. it is possible that larger flow values have the highest influence

Technical corrections: 7. P2/L22- Please clarify the "effects of past climate change" as the anthropogenic effects are mainly observed after the 20th century. Related to this, the next line indicates "high long-term variability" of reconstructions, which should be mainly representative of internal variability. 8. P2/L32: Please remove "that must be confronted" 9. P3/L2: Use either "streamflow" or "stream flow" throughout the paper 10. P3/L2: Is an R2 value of 0.76 low considering that it is based on an indirect estimate of streamflow? 11. P3/L23: What does "many uncertainties" imply? Large uncertainties

or many sources of uncertainties? If the latter, please provide a few others and add references 12. P5/L20: What are the chronology predictors? 13. P6/L9-10- Statement is not clear 14. P7/L6-7- I suggest rephrasing this statement for example "...smaller variability compared to...because of ..." 15. P8/L7- the frequency of what? 16. P8/L12- I think qstd should be in the denominator and qmean in the nominator 17. Table 1- Please spell out the predictors before or after the table. How was the predictor selection performed? and how did the authors consider multicollinearity? 18. P15/L4- In the introduction, it is mentioned that one of the challenges of current approaches is the low R2 values (0.37–0.76). It implied that this issue was addressed in the study, however current results are within this range. Please clarify

---

## Author Comment (AC1) · 12 Nov 2019

We are extremely grateful for the constructive interactive reviews of our paper entitled: "Paleo-hydrologic reconstruction of 400 years of past flows at a weekly time step for major rivers of Western Canada". The comments are constructive and will help immensely in improving the paper. The format of this author response will be to address each reviewer comment sequentially by first quoting the comment and then providing a response immediately below.

[Figure]

Question 1 "Why should we make sure that the reconstructed flows "properly preserve the statistical properties of the reference flows, particularly, short- to long-term persistence and the structure of variability across time scales", mentioned later in the abstract and used in the methodology"

This is an important comment, and the paper has been changed to make this point clearer. This point has been mainly ignored in the previous work, as discussed in Razavi et al. (2016) and Razavi and Vogel (2018). The reconstructed flows over 'the reference period' were constructed in a way to preserve the statistical properties of 'the reference flows'. This is to ensure that the important statistical properties in the entire reconstructed flow dataset are sufficiently representative. The outcome of any modelling study that uses these flows will depend to a large extent on the presence of these statistical properties. Many performance measures for example in a water management modelling study will depend on the variability of weekly flow or the long-term autocorrelation in the flow.

Question 2 "Similar to Razavi et al. (2016), only the four ..." needs to be explained more in-depth in this manuscript to make it self-contained."

The reviewer makes a good point by this comment relating to the MLR models for the North Saskatchewan River and the Oldman River sub-basins using the four and eight chronology sites falling within the sub-basins, respectively. It was thought that the MLR models would have less uncertainty if they were constructed using the chronology sites falling within the respective basins. Unfortunately, the Red Deer River and Bow River sub-basins contain few or no chronology sites, so as a way around this, all the chronology sites were used in the construction of their MLR models. As evident in the results shown in Figure 3, the use of chronology sites falling within a particular sub-basin to construct the MLR model for that sub-basin does in fact increase the accuracy of the reconstructed biennial flows. We will make sure that this point is clear in the revised manuscript.

Question 3 "Adding to comment 2 above, the following methods/approaches need to be explained in Section 2 to make the manuscript self-contained: a. Page 5: why MLR is expected to reconstruct flow from tree ring data, adequately. b. Page 5: The "leave-one-out cross-validation strategy" c. Page 6: The "random matching".

The reviewer brings up very good points by these comments, and addressing them in our paper should improve the description of the methodology considerably. Multiple linear regression (MLR) is a simple but effective method of modelling the direct monotonic (approximately linear) relationship between tree growth rate and flow. Since both flow and tree ring growth depend on soil moisture, we can expect a fairly linear relationship. The "leave-one-out-cross-validation-strategy" maximises the validation of the MLR models given that the overlap period between the tree-ring chronologies and reference flows is relatively short. Here, if the overlap period is n years, n-1 validations are possible by sequentially calibrating using n-1 years of data and validating against the year left out, with the average $R2$ of the n-1 validations used in the final analysis of the model performance. Random matching relates to step four in Figure 2, where as the reconstructed biennial flows are stepped through, a reference average biennial flow with similar hydrological properties is randomly selected. The randomness of the selection allows an ensemble approach to be adopted as multiple datasets can be generated that are different but retain the same underlying statistical properties. We will make sure that these points are clear in the revised manuscript.

Comment 4 "Authors need to elaborate on the negative regression coefficients in table 1. Are they physically meaningful? Could the regression be constrained to take only non-negative coefficients? In the same table, variables (e.g. WWP and JOLA) have to be introduced."

This is a very good point. The tree growth (as represented by three ring width) and water availability (as represented by streamflows here) are always positivity correlated in moisture-limited settings (not necessarily true in energy-limited settings). This means that a regression coefficient in a single linear regression should always be positive.

However, in multiple linear regression, the signs of some of the coefficients might become negative because of collinearity, which relates to the dependence of the tree-ring chronologies. The only way to avoid seeing negative signs is to apply principal component analysis (PCA) to the repressors before doing regression. However, applying PCA would not improve the predictive power of the regression in this case, and therefore, has not been conducted. We will make sure to explain this point in the revised manuscript. We will also define the variables in the regression equations more carefully.

Comment 5 "Page 7: What do author mean by naturally in "The biennial reconstructed time series naturally demonstrate smaller variability compared with the biennial flows in the reference period, when MLR models are used for reconstruction.""

The reviewer has highlighted the use of a term which may cause confusion, and the paper has been adapted to remove the use of the word "naturally". The MLR models fitted by the Least Square Method always produce smaller variance compared with the variance of observations. As such, reconstructed flows will have less variability compared to the reference flows as the tree-ring chronologies will not explain all the variation in the reference flow. The word "Naturally" has been replaced with "As expected".

Comment 6 "The two year instrumental periods briefly introduced in the abstract need to be explained more in-depth in Section 2. 7."

The reference to a "two-year instrumental period" relates to matching the broad properties of the biennial reconstructed flow with those of the biennial average reference flow. In this study, these properties were the hydrological category of wet, average or dry and whether the wetter year in the biennial average occurred in the first or second year. The paper has been adapted to make this point clearer.

Comment 7 "The persistence calculation should be explained in Section 2."

We appreciate the reviewer drawing attention to the need for further explanation. Fur-

ther description of the autocorrelation calculation has been added to Section 2.

---

## Author Response (AR1)

We are extremely grateful for the constructive interactive reviews of our paper entitled: "Paleo-hydrologic reconstruction of 400 years of past flows at a weekly time step for major rivers of Western Canada". The comments were constructive and helped immensely in improving the paper. The format of this author response is to address each reviewer comment sequentially by first quoting the comment and then providing a response immediately below.

Review 1:

Question 1

"*Why should we make sure that the reconstructed flows "properly preserve the statistical properties of the reference flows, particularly, short- to long-term persistence and the structure of variability across time scales", mentioned later in the abstract and used in the methodology*"

This is an important comment, and the paper has been changed to make this point clearer (Page 16, lines 4-7). This point has been mainly ignored in the previous work, as discussed in Razavi et al. (2016) and Razavi and Vogel (2018). The reconstructed flows over 'the reference period' were constructed in a way to preserve the statistical properties of 'the reference flows'. This is to ensure that the important statistical properties in the entire reconstructed flow dataset are sufficiently representative. The outcome of any modelling study that uses these flows will depend to a large extent on the presence of these statistical properties. Many performance measures for example in a water management modelling study will depend on the variability of weekly flow or the long-term autocorrelation in the flow.

Question 2

"*Similar to Razavi et al. (2016), only the four …" needs to be explained more in-depth in this manuscript to make it self-contained.*"

The reviewer makes a good point by this comment relating to the MLR models for the North Saskatchewan River and the Oldman River sub-basins using the four and eight chronology sites falling within the sub-basins, respectively. It was expected that the MLR models would have less uncertainty if they were constructed using the chronology sites falling within the respective basins. Unfortunately, the Red Deer River and Bow River sub-basins contain few or no chronology sites, so as a way around this, all the chronology sites were used in the construction of their MLR models. As evident in the results shown in Figure 3, the use of chronology sites falling within a particular sub-basin to construct the MLR model for that sub-basin does in fact increase the accuracy of the reconstructed biennial flows. This point has been made clearer in the manuscript (Pg 5 lines 20-25).

Question 3

"*Adding to comment 2 above, the following methods/approaches need to be explained in Section 2 to make the manuscript self-contained: a. Page 5: why MLR is expected to reconstruct flow from tree ring data, adequately. b. Page 5: The "leave-one-out cross-validation strategy" c. Page 6: The "random matching*".

The reviewer brings up very good points by these comments, and addressing them in our paper should improve the description of the methodology considerably. Multiple linear regression (MLR) is a simple but effective method of modelling the direct monotonic (approximately linear) relationship between

tree growth rate and flow. Since both flow and tree ring growth depend on soil moisture, we can expect a fairly linear relationship. This has been stated in the paper on Page 5 lines 15-20. The "leave-one-out-cross-validation-strategy" maximises the validation of the MLR models given that the overlap period between the tree-ring chronologies and reference flows is relatively short. Here, if the overlap period is n years, n-1 validations are possible by sequentially calibrating using n-1 years of data and validating against the year left out, with the average $R^2$ of the n-1 validations used in the final analysis of the model performance. This type of validation can also help identify and avoid overfitting. A short explanation has been included on Page 5, lines 28 – 30. Random matching relates to step four in Figure 2, where as the reconstructed biennial flows are stepped through, a reference average biennial flow with similar hydrological properties is randomly selected. The randomness of the selection allows an ensemble approach to be adopted as multiple datasets can be generated that are different but retain the same underlying statistical properties. This has been stated on Page 6, line 20

Comment 4

"*Authors need to elaborate on the negative regression coefficients in table 1. Are they physically meaningful? Could the regression be constrained to take only non-negative coefficients? In the same table, variables (e.g. WWP and JOLA) have to be introduced.*"

This is a very good point. The tree growth (as represented by three ring width) and water availability (as represented by streamflows here) are always positivity correlated in moisture-limited settings (not necessarily true in energy-limited settings). This means that a regression coefficient in a single linear regression should always be positive. However, in multiple linear regression, the signs of some of the coefficients might become negative because of collinearity, which relates to the dependence of the tree-ring chronologies. One way to avoid seeing negative signs is to apply principal component analysis (PCA) to the repressors before doing regression. However, applying PCA would not improve the predictive power of the regression in this case, and therefore, has not been conducted. This point has been included in the manuscript (Page 15 lines 8-16). The variables within the MLP models such as 'WWP' and 'JOLA' are codes for tree-ring chronology sites, and we have included a citation where these codes are defined (Page 9 in Table 1).

Comment 5

"*Page 7: What do author mean by naturally in "The biennial reconstructed time series naturally demonstrate smaller variability compared with the biennial flows in the reference period, when MLR models are used for reconstruction."*

The reviewer has highlighted the use of a term which may cause confusion, and the paper has been adapted to remove the use of the word "naturally". The MLR models fitted by the Least Square Method always produce smaller variance compared with the variance of observations. As such, reconstructed flows will have less variability compared to the reference flows as the tree-ring chronologies will not explain all the variation in the reference flow. The word "Naturally" has been replaced with "As expected" and this point has been explained in the paper (Bottom of page 7).

Comment 6

"*The two year instrumental periods briefly introduced in the abstract need to be explained more in-depth in Section 2. 7.*"

The reference to a "two-year instrumental period" relates to matching the broad properties of the biennial reconstructed flow with those of the biennial average reference flow. In this study, these properties were the hydrological category of wet, average or dry and whether the wetter year in the biennial average occurred in the first or second year. The paper has been adapted to make this point clearer (Page 6 line 8).

Comment 7

"*The persistence calculation should be explained in Section 2.*"

We appreciate the reviewer drawing attention to the need for further explanation. Further description of the autocorrelation calculation has been added to Section 2 (Page 8 line 21-25).

Reviewer 2

Comment 1

"*The authors use R2 to select the best MLR model. Did they consider possible over-parameterization and multicollinearity? Analyzing the standard errors and metrics such as information criteria (e.g. AIC) can identify the optimal models more reliably.*"

We first would like to thank the reviewer very much for his/her time and the constructive comments. The selection of the number of regressors are based on a former study with the same data (Razavi et al. 2016) where Akaike Information Criterion was used. This limited the risk of over-parametrization and over-fitting. Of course, there is always some level of collinearity among the tree ring chronologies. We note that we also applied principal component analysis (PCA) to the tree-ring chronologies and compared the results of MLR with and without PCA (this comparison is not reported), but didn't find any noticeable difference. This point has been made in the paper (Page 16, lines 9-13).

Comment 2

"*It is mentioned that "weekly flow distribution of the selected reference flow period can be used to construct the weekly flows". By doing so, the variability in the reconstructed flow will be similar to the reference flow variability, correct? Please discuss how/if this assumption can undermine the estimated variability of reconstructed flows? How does it account for possible recent trends due to the anthropogenic climate change effects?*"

The reviewer brings up an important point with this comment. Given the nature of annual tree growth rings, the intra-year variability of flows cannot be determined from tree-ring reconstructed flows. It is assumed that the reconstructed weekly flows will contain the long-term inter-year variability of flows as represented in the tree-ring data, but intra-year variability will be based on the variability of the reference flow, scaled to the biennial reconstructed flow. From a water resources modelling perspective, this longer-term variability is arguably more important to represent than intra-year variability; for example, it would be persistent long-term droughts that challenge the robustness of water resource systems. This point has been included in the paper (Page 17, lines 19-26)

Comment 3

*"The study focuses on matching the variation and persistence between the observed flow in the reference period and the reconstructed flow. I wonder whether historical/ recent physical processes could have distorted this similarity? For example, recent climate change trends are much stronger compared to the historical periods (prior to the reference period)"*

The reviewer's comment here highlights an important potential confounding factor in the analysis. Indeed, with increasing snow melt under a warming climate, the underlying statistical properties of flow in a water tower-driven catchment such as the Saskatchewan River basin may indeed change considerably compared to the pre-reference period. However, since the reference period used ended in 2000, which is now almost 20 years ago, we must assume that the more recent climate change trends were absent or at least much weaker prior to the turn of the century. The paper has been adapted to highlight this important point (Page 18, last paragraph).

Comment 4

*"P5/L8: I suggest discussing briefly the cause(s) for persistence in tree-ring chronologies and based on that justify why the multi-year approach will overcome the problem. Related to this, how about the role of teleconnection signals, which can affect the records over multiple years."*

The persistence in the tree-ring chronologies is a mixture of the persistence in the climate signal and the biological carry-over effects of trees. The latter refers to the fact that, for example, trees can tolerate water shortage in a dry year and grow well if the preceding year was wet. This effect diminishes for a longer time scale. The study of Razavi et al. (2016) has showed this.

The teleconnection signals typically occur in concert with precipitation in the region, and thus with water availability for tree growth. Therefore, tree-ring chronologies and their respective streamflow reconstructions should carry the teleconnection signals. We have included some discussion of these points in the paper (Page 5, lines 10-15).

.

Comment 5

*"P6/L15-16- It matches the first moment (i.e. the mean). How about higher moments like the variance?"*

The reviewer is referring to this sentence specifically: "The weekly distribution of flows in the selected biennial reference flow period is then used to construct the weekly flow reconstruction scaled to have the same biennial average as the original reconstructed biennial flow." The reviewer brings up an important point with this comment. As shown in Fig. 5 and Fig. 6, the disaggregation approach used was successful in replicating the variance in the reference flow within the reconstructed flow. The variance of the weekly reconstructed flow was in fact assessed during the study to be similar to that of the reference flow. The paper has been adapted to make this finding more explicit (Page 6 lines 26-28).

Comment 6

*"Figure 2, step 3- I wonder how much the yearly average is affected by seasonal variations? i.e. it is possible that larger flow values have the highest influence?"*

The reviewer makes an important point with this comment. At this point in the disaggregation, a relationship between the reconstructed flows and reference flows at a biennial scale is being made. Therefore, the seasonal variation at this point is not considered. We note that three rings provide no meaningful information on sub-annual variability. However, the influence of these much larger flows can be carried into the reconstructed weekly flows during step 4 in Figure 2. This point has been made in the paper (Page 16, second-last paragraph)

Comment 7 onwards: "Technical corrections"

*"P2/L22- Please clarify the "effects of past climate change" as the anthropogenic effects are mainly observed after the 20th century. Related to this, the next line indicates "high long-term variability" of reconstructions, which should be mainly representative of internal variability".*

Earth has always experienced strong climate change effects in the past unrelated to anthropogenic effects. Of course these effects have been intensified in the Anthropocene. Please refer to for example to Cohn and Lins (2005) and Razavi et al. (2015). The document has been adapted to include this point (Page 2, lines 23-25)

Cohn, T. A., & Lins, H. F. (2005). Nature's style: Naturally trendy. Geophysical research letters, 32(23).

Comment 8

*"P2/L32: Please remove "that must be confronted"*

We thank the reviewer for this comment. We have adapted the paper as requested.

Comment 9
*"P3/L2: Use either "streamflow" or "stream flow" throughout the paper."*

The paper has now been corrected in this regard.

Comment 10

*"P3/L2: Is an R2 value of 0.76 low considering that it is based on an indirect estimate of streamflow?"*

We thank the reviewer for bringing up this point. The sentence has been adapted to reflect mostly lower R2 values but some relatively strong relationships.

Comment 11

*"P3/L23: What does "many uncertainties" imply? Large uncertainties or many sources of uncertainties? If the latter, please provide a few others and add references."*

We implied the latter, and the manuscript will be adapted to include examples and citations.

Comment 12
*"P5/L20: What are the chronology predictors?"*

These refer to tree-ring stations used in the MLR models. The paper was adapted to make this clearer (Page 5, line 29).

Comment 13

*"P6/L9-10- Statement is not clear"*

We will ensure that this statement is re-written for improved clarity (Page 6, lines 19-23).

Comment 14

*"P7/L6-7- I suggest rephrasing this statement for example "...smaller variability compared to...because of ..."*

We appreciate the reviewer's suggestion and the paper will be corrected in this regard.

Comment 15

*"P8/L7- the frequency of what?"*

The reviewer highlights an error on our part. Since this is a flow duration curve and nota frequency curve, we should refer to duration and not frequency. We will correct this error in the paper.

Comment 16

*"P8/L12- I think qstd should be in the denominator and qmean in the nominator"*

We will double check whether this equation is written correctly in the manuscript.

Comment 17

*"Table 1- Please spell out the predictors before or after the table. How was the predictor selection performed? and how did the authors consider multicollinearity?"*

The predictors are codes for tree-ring chronology sites. We have provided a citation where these codes are defined. The selection of the number of regressors was based on a former study with the same data (Razavi et al. 2016) that used the Akaike Information Criterion, thereby limiting the risk of over-parametrization and over-fitting. We also applied PCA to the tree-ring chronologies and compared the results of MLR with and without PCA, but didn't find any noticeable difference.

Comment 18

"*P15/L4- In the introduction, it is mentioned that one of the challenges of current approaches is the low R2 values (0.37–0.76). It implied that this issue was addressed in the study, however current results are within this range. Please clarify*"

We thank the reviewer for highlighting this point. While the biennial reconstructions resulted in higher R2 than annual reconstructions for this region, our study did not merely aim to improve the regression fits within MLR models describing the relationships between tree-ring chronologies and naturalised flow. Rather, the paper introduces a method that firstly constructs these relationships between tree-ring chronologies and naturalised flow in a way that preserves persistence properties and variability of hydrological time series. Further, the study introduces a novel method of disaggregating biennial reconstructed flow to weekly flows. The uncertainty, firstly in the relationships between tree-ring chronologies and naturalised flow, and secondly within the disaggregation technique, is addressed through an ensemble approach, by producing a range of viable MLR models for individual catchments and multiple plausible flow time series within the disaggregation. The paper has been adapted to make this point clearer (Page 16, lines 7-14).

[revised manuscript text omitted]

The present study presents a novel approach for reconstructing flows from tree-ring data and disaggregating these flows to weekly while taking into account reconstruction uncertainty through an ensemble approach. As a result, four hundred years of weekly flows were generated for the four major sub-basins of the Saskatchewan River Basin. The Saskatchewan River is a river of great ecological, social and economic importance in Western Canada. The approach of reconstructing flow from tree-ring chronologies contains multiple sources of uncertainty, including the choice of predictor chronologies and the choice of disaggregation technique (Razavi et al., 2016). The present study presents an ensemble approach for encompassing these uncertainties.

**2 Case study, data and methodology**

**2.1 Case study**

The Saskatchewan River Basin (SaskRB) in Western Canada is a large river basin with an area of approximately 400,000 km² that transcends the Alberta, Saskatchewan and Manitoba provinces, and also extends into a small

part of the American State of Montana (Fig. 1). The east-facing Rocky Mountains to the west at over 3,000 m in elevation act as the 'water tower' of the basin (Martz et al., 2007; Pomeroy et al., 2005), contributing up to 95% of the total basin flow, after which the elevation of the basin drops to 750 m, 450 m and 300 m on the plains of Alberta, Saskatchewan and Manitoba, respectively, transitioning from alpine forest to prairie grasslands and river valley forest. Long sunny winters and short hot
5   summers characterise the climate of the basin, and there is a dramatic rainfall gradient from west to east, with precipitation of up to 1,500 mm year$^{-1}$ in the mountains to 300 mm year$^{-1}$–500 mm year$^{-1}$ on the semi-arid plains.

**2.2 Data**

Tree-ringTree-ring data for the major headwater tributaries of the SaskRB were used, namely the North Saskatchewan, Red Deer, Bow and Oldman rivers. Naturalised flows generated by Alberta Environment and Sustainable Resource Development
10   for four gauging stations (1912–2001) representing each of the four headwater tributaries were used in the analysis (see Fig. 1). A total of 16 tree-ringtree-ring chronology sites were used, with most from sites chosen for soil moisture availability only during snowmelt or rainfall. These chronology data were obtained from the Prairie Adaptation Research Collaborative (PARC; www.parc.ca). Razavi et al. (2016) describe the tree-ringtree-ring measurement, detrending and averaging procedures used. The tree-ringtree-ring chronologies used represent tree growth rates from 1600 to 2001.

[Figure]

Figure 1. Map of the tributaries of the Saskatchewan River Basin and the locations of the chronology sites (circles) and streamflow gauges (cones)

**2.3 Methodology**

**2.3.1 Reconstruction of flows based on tree-ring chronologies**

The traditional method of reconstructing streamflow based on tree-ring chronologies is through correlations between tree-ring chronologies and observed streamflow at the yearly scale. However, as shown by Razavi et al. (2016), there are stronger correlations between tree-ring chronologies and streamflow at multi-year time scales. In addition, establishing correlations at multi-year time scales can be a viable method of overcoming the significantly higher persistence in tree-ring chronologies compared to streamflow without resorting to prewhitening techniques, which have been shown to result in a loss of information not related to autocorrelation (Razavi et al., 2016; Razavi and Vogel, 2017). The

persistence present in tree-ring chronologies is due to a mixture of the persistence in the climate signal and the biological carry-over effects of trees, which for example, would allow trees to tolerate a water shortage in a dry year if the preceding year was wet. In addition, teleconnection signals typically occur in concert with precipitation in a region, and thus with water availability for tree growth. Therefore, tree-ring chronologies and their respective streamflow reconstructions should carry the teleconnection signals. As shown by Razavi et al. (2016), the persistence effect diminishes over a longer time scale. Although the relationship between streamflow and tree-ringtree-ring chronologies strengthens at multi-year time scales, establishing this relationship at an optimal timescale that can be in the order of 5 years (Razavi et al., 2016) would disadvantage the process of disaggregation of deconstructed reconstructed flow. Therefore, as a compromise, the present study established statistical relationships between two-year moving averages of tree-ringtree-ring chronologies and naturalised streamflows. Multiple linear regression (MLR) fitted by least squares was the statistical approach taken to establish the relationships between tree-ringtree-ring chronologies and streamflows. We can expect a fairly linear relationship between flow and tree-ring growth as observed in many regionssince both depend on soil moisture of the world (Axelson et al., 2009; Boucher et al., 2011; Case & MacDonald, 2003; Gou et al., 2007; Souchyn & Ilich, 2017)(REF), and (MLR) is a simple but effective method of representing this direct monotonic (approximately linear) relationship between tree growth rate and flow. Since both flow and tree-ring growth depend on soil moisture, we can expect a fairly linear relationship. The predictive ability of the models was assessed through the coefficient of determination, $R^2$. Similar to Razavi et al. (2016), since the North Saskatchewan River and Oldman River sub-basins contained sufficient chronology sites, only the four and eight chronology sites falling within the sub-basins of the North Saskatchewan River sub-basin and Oldman River sub-basin, respectively were used to establish their respective MLR models;, whereas since few or no sites were present in the Red Deer River and Bow River sub-basins (Fig. 1), all 16 chronologies were used to establish their MLR models for the Red Deer River and Bow River sub-basins (Fig. 1). Building on the experience of Razavi et al. (2016) and employing the Akaike Information Criterion, models with both three and two chronology predictors (tree-ring stations) were established for the North Saskatchewan River and Oldman River sub-basins, whereas MLR models with only two chronology predictors were established for the Bow River and Red Deer River sub-basins. MLR models were generated for the shared period between tree-ringtree-ring chronologies and naturalised streamflows of 1912–2001. A leave-one-out cross-validation strategy was used to test the performance of each model. The is validation-strategy maximises our ability in validatingthe validation of the MLR models using the relatively short overlap period between the tree-ring chronologies and reference flow and provides a more accurate measure of model goodness of fit and can identify and avoid overfitting. To account for the uncertainty in streamflow reconstructions, multiple MLR models with the best performances were selected for each sub-basin. The choice of the best models for each sub-basin was rather subjective, and depended on relative $R^2$ values achieved for all the models generated.

**2.3.2 Disaggregation of two-year reconstructed flows to weekly**

Although there are definite advantages in reconstructing streamflow from tree-ringtree-ring chronologies at multi-year time scales (Razavi et al., 2016), the usefulness of these flows for evaluating water resources systems is limited. Generating weekly

reconstructed streamflow would be of more use in evaluating water resources systems as, for example, the established water management model, the Water Resources Management Model (WRMM) (Alberta Environment, 2002) used within the Prairie Provinces, runs on a weekly time step. The conceptual approach taken can be represented in Fig. 2. The basic premise adopted is that biennial reconstructed flow can be disaggregated to weekly reconstructed flow by the selection of biennial flow periods

5    from the reference naturalised flow (1912–2001) (two-year instrumental periods) with similar attributes to the biennial reconstructed flow, after which the weekly flow distribution of the selected reference flow period can be used to construct the weekly flows. The attributes used to match the biennial reconstructed flow with biennial reference flow were hydrological condition, simply defined as dry, normal and wet conditions corresponding to flow less than the 25$^{th}$ percentile, between the 25$^{th}$ percentile and the 75$^{th}$ percentile, and greater than the 75$^{th}$ percentile, respectively, and which year in each biennial flow,

10   year 1 or year 2, contributes the greater amount of flow. In Fig. 2, (1) represents the average tree-ring growth rates of all the tree-ring chronologies used to generate a particular MLR model for a sub-basin on a yearly time step. By examining the yearly tree-ring growth rates in pairs (on a biennial scale), it can be determined which yearly growth rate, that of year 1 or year 2, is larger. And since we constructed biennial flows from biennial tree-ring growth rates, tis allows B for each biennial flow in (2) to be set to 1 or 2 to indicate whether the first or second year of that biennial flow value contributed

15   the greater flow. 'A' in (2) represents the hydrological condition explained earlier. A similar process is performed for the weekly naturalised reference flow in (3), except that average yearly flows are used to set 'B'. The biennial reconstructed flow is then stepped through in (4), and a similar period according to A and B is randomly selected from the biennial reference flow. The approach of random matching allows an ensemble of weekly flow reconstructions to be generated for each single biennial flow reconstruction while retaining the same underlying statistical properties. The weekly distribution of flows in the selected

20   biennial reference flow period is then used to construct the weekly flow reconstruction, scaled to have the same biennial average as the original reconstructed biennial flow. An  argument could be made that the scaling should be according to the variance of the biennial reference flow; however, the variance of the reconstructed flow was found to be similar to that of the reference flow using this approach.

[Figure]

(1) Read in the average yearly tree-ring growth rates (GRs) used to generate the MLR model of reconstructed biennial flow.

(2) Read in the biennial reconstructed flow generated through the MLR model and set A and B, where A = 'Dry', 'Normal' or 'Wet' if the flow is less than the 25th percentile, between the 25th percentile and the 75th percentile, and greater than the 75th percentile, respectively, and B = 1 or 2 according to the greater of the corresponding annual GRs from (1), e.g. for 1600 and 1601 in (1), 1601 has the larger growth rate, therefore B is set to 2 for 1600 in (2).

(3) Read in the weekly naturalised flow and generate a second time series of yearly average flow. Then generate a third time series of biennial average flow. Set A and B, where A = 'Dry', 'Normal' or 'Wet' if the flow is less than the 25th percentile, between the 25th percentile and the 75th percentile, and greater than the 75th percentile, respectively, and B = 1 or 2 according to the greater of the corresponding average yearly flow.

(4) Step through reconstructed biennial flow and randomly identify a reference average biennial flow with the same values for A and B. Reconstruct the weekly flow using the weekly distribution of the reference weekly flow for the corresponding years.

(5) The disaggregation process can result in some loss of variation in the flows at a yearly scale. A scaling process was implemented to scale the flow duration curve (FDC) of the of the reconstructed yearly flow to have the same shape as that of the reference yearly flow. Statistical scaling was implemented to ensure the yearly reconstructed flows have the same standard deviation as the reference yearly flows.

Figure 2. Conceptual representation of the process for disaggregating two-year (biennial) tree-ring reconstructed streamflow to weekly reconstructed streamflow

As expected, the biennial reconstructed time series  demonstrate smaller variability compared with the biennial flows in the reference period when MLR models are used for reconstruction. as the MLR models fitted by the least square

method always produce smaller variance compared with the variance of observations. Therefore, the resulting annual and weekly time series also have less variability compared with their counterparts in the reference period. To rectify this problem, the reconstructed flows generated by stage (4) in Fig. 2 over the reference period (1912–2001) were compared to the reference flow at a yearly scale in the form of flow duration curves (FDCs). Typically, loss of variance in the reconstructed flows will

5 manifest as fewer extreme high and low flows. A scaling equation was implemented to scale the FDC of the reconstructed flow to have the same shape as that of the reference flow:

$$q' = q \times (A \times P^B + C),$$ (Eq. 1)

where $q'$ is the scaled yearly reconstructed flow (m³ s⁻¹), $q$ is the yearly reconstructed flow (m³ s⁻¹), $P$ is the  duration (%) and $A$, $B$ and $C$ are parameters that are calibrated by fitting the scaled yearly flow reconstructions for the reference period

10 (1912–2001) to the yearly reference flow FDC.

In addition, the scaled yearly reconstructed flows were re-scaled according to the mean and standard deviation of the yearly reference flows:

$$q'' = \frac{q' - q'_{stdev}}{q'_{mean}} \times Q_{stdev} + Q_{mean} ,$$ (Eq. 2)

where $q'_{stdev}$ and $q'_{mean}$ are the standard deviation and mean of the scaled yearly reconstructed flow for the reference period

15 (1912–2001), respectively, $Q_{stdev}$ and $Q_{mean}$ are the standard deviation and mean of the yearly reference flow, respectively and $q''$ is the final (re-scaled) yearly reconstructed flow for the entire reconstruction period (1600–2001).

This update of the yearly reconstructed flows was used to scale the weekly flow reconstructions, which were in turn scaled to have the same biennial average as the original biennial flow reconstructions.

**2.3.3 Comparisons of autocorrelation between weekly flow reconstructions and reference flow**

Weekly reference flows and weekly reconstructed flows were averaged to yearly over the reference period (1912–2001) and autocorrelation was calculated for different yearly time lags from one to ten years. This was performed to confirm that the important statistical property of autocorrelation in the reference flows was carried over into the reconstructed flows.

**3 Results**

**3.1 Reconstruction of biennial flows based on tree-ring chronologies**

MLR models with the best $R^2$ values for each sub-basin were chosen to reconstruct biennial flows based on tree-ring chronologies (Table 1). Four, nine, six and ten MLR models were chosen for the North Saskatchewan, Oldman, Red Deer and

30 Bow sub-basins, respectively, with $R^2$ values ranging from 0.50–0.56, 0.44–0.51, 0.45–0.55 and 0.49–0.56, respectively.

Figure 3 shows the time series of reconstructed two-year (biennial) flows for all the MLR models shown in Table 1 for the four sub-basins along with the reference flow over the calibration period.

Table 1. Regression equations and $R^2$ values obtained in two-year (biennial) flow reconstructions using tree-ring chronologies for the Saskatchewan River Basin. Tree-ring chronology site code definitions can be found in Sauchyn et al. (2011).

[revised manuscript text omitted]

While the MLR $R^2$ values obtained in the current study are within the range of those of previous studies describing the relationship between tree-ring chronologies and river flow, the present study did not focus merely to improve the regression fitsInstead, it introduced a method that firstly constructs these relationships between tree-ring chronologies and naturalised flow in a way that preserves persistence properties and variability of hydrological time series, and secondly introduced a novel method of disaggregating biennial reconstructed flow to weekly flows. The uncertainty, firstly in the relationships between tree-ring chronologies and naturalised flow, and secondly within the disaggregation technique, is addressed through an ensemble approach, by producing a range of viable MLR models for individual catchments and multiple plausible flow time series within the disaggregation.

The selection of the number of regressors  was based on a former study with the same data (Razavi et al., 2016) where the Akaike Information Criterion was used to limit the risk of over-parametrization and over-fitting.

The tree growth and water availability, as represented by tree-ring width and streamflows here, respectively, are always positivity correlated in moisture-limited settings (although this is not necessarily true in energy-limited settings). This means that a regression coefficient in a single linear regression should always be positive. However, as evident by the negative coefficients of the MLR models shown in Table 1, in multiple linear regression, the signs of some of the coefficients might become negative because of collinearity, which relates to the dependence of the tree-ring chronologies.  The application of principal component analysis (PCA) to the regressors before doing regression is the way  of circumventing collinearity and avoiding negative coefficients in the models.  Although not reported here, PCA was applied to the tree-ring chronologies and results of MLR with and without PCA were compared, with no noticeable difference found.

5   We note that tree rings provide no meaningful information on sub-annual variability.  Therefore, the process of disaggregating two-year (biennial) tree-ring reconstructed streamflow to weekly reconstructed streamflow  is merely statistical, based on the information on the seasonality in the reference record.
10   As such, a caveat of this approach is the absence of information on any possible change in seasonality due to climate change effects.
15   Arguably though, from a water resources modelling perspective, the longer-term inter-year variability is more important to represent than intra-year variability; for example, it would be persistent long-term droughts that challenge the robustness of water resource systems. The discussion below shows that statistical properties of the reference flows are preserved in the reconstructed flows. This is important as the outcomes of modelling studies that would use these flows will depend largely on
20  the presence of these statistical properties.

The approaches of using the weekly distribution of the reference flow within the disaggregation of biennial reconstructed flows
25  [see Fig. 2 step (4)] along with the scaling of the yearly reconstructed flow [see Fig. 2 step (5)] appear to resolve the issue of discrepancies in variation and persistence between the reconstructed and reference flow. Figure 5 shows that the weekly reconstructed flow displays the same timing and range of flow in comparison to the reference flow, and also similar timing of dry and wet periods. Figure 5 and Fig. 6 show that the disaggregation approach used was successful in replicating the variance in the reference flow within the reconstructed flow. The persistence between flows for both the reference and reconstructed
30  flows were similar, and both showed a generally decreasing trend with increasing time lag (Fig. 7).

Razavi et al. (2016) showed that tree-ring chronologies and flows at the annual time scale may possess inconsistent persistence properties. This inconsistency leads to dissimilar patterns of change and variability in the two types of time series across other time scales, which might invalidate any resulting flow reconstructions. To investigate this,

 Figure 8 shows the variance of the reconstructed and reference flows at different time scales on a log–log scale for the four sub-basins. The variance values at different time scales were calculated through averaging, so for example, a flow period of 100 years would yield 50 and ten values when the average of every two and ten years is calculated, respectively. The graph represents variance of the different time series over different time scales. The slopes of the different time series can be benchmarked against a random process (the red dotted line in the plot) which contains no persistence at any time scale. The differences in slopes between the reconstructed and reference flows shown in Fig. 8 compared to that of the random process can be attributed to persistence at the range of time scales represented. Razavi et al. (2016) using a similar plot showed that tree-ring growth rates have considerably different persistence at shorter time scales compared to flow, and this persistence could be expected to be transferred to flow reconstructed from tree-ring chronologies using relationships established at shorter time scales. The slope associated with reference flow would however be closer to that of the random process at a shorter time scale. Figure 8 demonstrates that the flow reconstruction and disaggregation method used in the present study appears to overcome the problem of transferal of higher persistence in tree-ring chronologies to the reconstructed flow at shorter time scales. The slopes of both the reference and reconstructed flows appear to be similar . This indicates that the reconstructed flows have similar persistence to the reference flows across the entire reference flow period of 90 years. Variance versus time scale plots provide a method of studying the 'Hurst Phenomenon' for long-term persistence in hydrologic time series (Hurst, 1951).

~~The results show that statistical properties of the reference flows are preserved in the reconstructed flows. This is important as the outcomes of modelling studies that would use these flows will depend largely on the presence of these statistical properties. A good argument can be made that this approach may undermine the estimated intra-year variability of reconstructed flows, particularly within the context of climate change. However, the intra-year variability of flows cannot be determined from tree-ring reconstructed flows, and rather the reconstructed weekly flows will contain the long-term inter-year variability of flows as represented in the tree-ring data. Arguably, from a water resources modelling perspective, this longer-term variability is more important to represent than intra-year variability; for example, it would be persistent long-term droughts that challenge the robustness of water resource systems.~~

[Figure]

Figure 8. Variance-versus-time-scale plot for reference and reconstructed flows in the (a) North Saskatchewan, (b) Oldman, (c) Red Deer, and (d) Bow Rivers. Black and blue time series represent reference and reconstructed flows, respectively, whereas the red line represents a random process with no long-term persistence

The present study reconstructed biennial flows based on MLR models of the relationship between tree-ring chronologies and observed flow in the reference period, and an  argument can be made that changes in recent physical processes due to climate change could have distorted this relationship. Indeed, with increasing snow melt under a warming climate, the underlying statistical properties of flow in a water tower-driven catchment such as the SaskRB may indeed change considerably compared to the pre-reference period. However, since the reference period used ended in 2000, which is now almost 20 years ago, we  assume that the more recent climate change trends were absent or at least much weaker prior to the turn of the century.

[revised manuscript text omitted]